# A New Mini-Invasive Approach for a Catastrophic Disease: Staged Endovascular and Endoscopic Treatment of Aorto-Esophageal Fistulas

**DOI:** 10.3390/jpm12101735

**Published:** 2022-10-19

**Authors:** Federica Donato, Ivo Boskoski, Claudio Vincenzoni, Francesca Montanari, Giovanni Tinelli, Tommaso Donati, Yamume Tshomba

**Affiliations:** 1Unit of Vascular Surgery, Fondazione Policlinico Universitario Agostino Gemelli IRCCS-Università Cattolica del Sacro Cuore, 00168 Rome, Italy; 2Digestive Endoscopy Unit, Fondazione Policlinico Universitario Agostino Gemelli IRCCS-Università Cattolica del Sacro Cuore, 00168 Rome, Italy

**Keywords:** aorto-esophageal fistula, TEVAR, aneurysm

## Abstract

Aorto-esophageal fistula (AEF) is an uncommon but usually fatal disorder. Surgery with resection of an aneurysm and esophagus, in situ reconstruction of the descending aorta and omental flap installation offers the gold standard for the reduction of infections, but it is burdened by high intraoperative and perioperative mortality rates. We report our experience with a combined minimally invasive approach for the multi-stage treatment of three cases of aorto-esophageal fistula caused by thoracic aneurysm rupture. In all of the patients, the aneurysm was treated with thoracic endovascular aortic repair and the esophageal lesion was treated with esophageal endoprosthesis placement. According to our experience, the combined strategy of thoracic endovascular aortic repair (TEVAR) and esophageal less invasive endoscopic treatments represents an alternative solution in frail patients with high surgical risk.

## 1. Introduction

Aorto-esophageal fistula (AEF) is an uncommon but usually fatal disorder. The most frequent causes of AEF are ruptured or non-ruptured aortic aneurysm, foreign body ingestion, trauma and advanced esophageal carcinoma [1]. Secondary AEFs can occur after aortic reconstructive surgery with prosthetic grafts or thoracic endovascular aortic repair (TEVAR) [2,3]. The classic symptoms of AEF involve Chiari’s triad of mid-thoracic pain, sentinel arterial hemorrhage and final massive hematemesis after a symptom-free interval [1]. 

We report the combined minimally invasive approach for the multi-stage treatment of three cases of AEF following the rupture of thoracic aneurysm. 

## 2. Case Report

### 2.1. Case 1

A 79-year-old male patient presented with hematemesis. The patient had a long history of smoking, hypertension, diabetes mellitus, critical limb ischemia with foot ulcers and HCV-related liver disease. At admission, the patient was conscious, oriented and hemodynamically stable. Laboratory tests revealed hemoglobin 11 g/dL, white blood cells 16,000 cells/mm^3^ and C-reactive protein 21.23 mg/dL. The urgent computed tomography (CT) scan showed mediastinal collection (74 × 30 mm) anterior to the thoracic aorta and close to distal esophagus with irregularity of the left anterolateral wall of the aortic bleb, suggestive of a ruptured thoracic aortic aneurysm with aorto-esophageal fistula. TEVAR was performed in emergency with a technical success. The patient recovered well, and an esophagogastroduodenoscopy (EGD) was performed one week later and confirmed the presence of 10 mm AEF at 40 cm beyond the dental arch. Abundant pus was coming out from the fistulous orifice; therefore, two double pigtail plastic stents (8F diameter, 20 mm length) were placed through the fistula (Figure 1). Five days after the EGD, a jejunostomy was placed for enteral nutrition. Intravenous empiric antibiotic therapy with fluconazole, piperacillin/tazobactam and linezolid was initiated. The postoperative course was uneventful. At one month, a new CT scan was performed and showed complete reduction of the purulent collection (Figure 2). The jejunostomy was removed, with resumption of oral nutrition. One month later, the two double pigtail stents were removed endoscopically. Six months from the index procedure, an FDG PET-TC and a sallow study with gastrografin showed no signs of infections and no leakage (Figure 3). At the time of writing of this paper, the patient was asymptomatic and on a free diet.

### 2.2. Case 2

A 75-year-old male patient was admitted in the emergency department after a violent episode of hematemesis. The patient had a long history of smoking, hypertension, rectal carcinoma treated with resection and permanent colostomy. On admission, the patient was conscious but hemodynamically unstable. Urgent CT and esophagogastroduodenoscopy showed a ruptured thoracic aortic aneurysm and a fistula between the aorta and esophagus. A TEVAR was performed in proximal landing zone 0, using a bilateral percutaneous femoral approach. Simultaneously, a total debranching of the epiaortic vessels by carotid-carotid-subclavian bypass was performed. The patient recovered well, and an EGD performed five days later confirmed the presence of a 9 mm AEF at 36 cm beyond the dental arch with pus leaking from the fistulous orifice. Two double-tailed plastic stents (8F diameter, 20 mm length) were placed across the fistula (Figure 4). Three days after EGD, a jejunostomy was placed for enteral nutrition. Broad-spectrum empiric intravenous antibiotic therapy and antifungal therapy was initiated. The postoperative course was uneventful. At 1 month, a new CT scan was performed that showed complete reduction of the purulent collection. The jejunostomy was removed, with resumption of oral nutrition. Two months later, the two double-tailed stents were removed endoscopically. 

Six months after the index surgery, an CT scan and a gastrographin swallowing study showed no signs of infection and no leakage.

### 2.3. Case 3

A 67-year-old male patient was admitted in the emergency department for acute mid-thoracic pain. His past medical history included hypertension, ankylosing spondylitis, HCV-related liver disease, closure of a patent foramen ovale and left hip prosthesis.

An emergency CT scan showed AEF, and emergency TEVAR was performed. A control CT scan at two days showed expansion and weakening of the walls of the distal esophagus with continuous solution along the lateral wall. Upper gastrointestinal endoscopy confirmed the presence of 10 mm AEF located in the distal esophagus (from 34 cm to 40 cm from the incisors). The aortic prosthesis was visible endoscopically in the esophagus. Three double 8F pigtail plastic stents were placed through the fistula (Figure 5). A jejunostomy was placed for enteral nutrition and intravenous empiric antibiotic therapy was started.

The postoperative course was uneventful. At one month, a new CT scan was performed and showed reduction of the purulent collection and the jejunostomy was removed.

At three months, the patient underwent an intraoperative endoscopy which showed complete closure of the esophageal lesion (Figure 6) and the two double pigtail stents were removed endoscopically.

A control CT scan and FDG PET-CT at 6 months after the index procedure showed no collections.

## 3. Discussion

AEF remains a life-threatening condition with a high rate of morbidity and mortality; 54.2% of cases are secondary to rupture of a descending thoracic aorta aneurysm into the esophagus [1,4].

Conservative medical treatment has a poor prognosis and invariably results in a fatal outcome [1,5].

The management of AEF includes the bleeding control, the aorta and esophagus treatment and the infection control. Despite several treatment strategies, surgery with resection of an aneurysm and esophagus, in situ reconstruction of the descending aorta and omental flap installation offers the gold standard for the reduction of infections [6,7]. Open surgical access allows the control of both the aneurysm and the esophagus as well as the drainage of the mediastinum. However, this approach is commonly performed via a left posterolateral thoracotomy, and it is burdened by high intraoperative and perioperative mortality rates.

TEVAR has become a widely accepted intervention for AEFs, and it may be an alternative to open surgery in the emergent setting or in high-risk elderly patients [8,9,10].

Although most previous reports recommend TEVAR as a bridging treatment to control bleeding [8,11], some authors suggest considering TEVAR as a potential aortic definitive treatment in patients at high risk for open surgical repair and without clinical evidence of infection [12,13,14].

The esophageal repair can be performed with several techniques.

Radical esophagectomy is usually the most effective strategy to treat esophageal defects [6,11], but it increases surgical trauma and the rate of perioperative death. Bipolar esophageal exclusion may be a valid alternative to esophageal resection. Direct suture of small defects of the esophageal wall has become possible in selected cases with no significant mediastinitis [14,15].

The recent technological development allowed endoscopic therapy to get a primary role in the management of AEFs. Esophageal stents or esophageal clips and suturing are minimally invasive alternatives to surgical reconstruction, and they can prevent aortic graft contamination, which is an important factor in long-term prognosis.

In our experience, the therapeutic choices were influenced by the poor physical state of the patients.

A multi-stage treatment was planned to improve results and allow for an adequate nutritional supply and intravenous antibiotic therapy.

In our patients, TEVAR represented a minimally invasive procedure and allowed us to reduce intraoperative and perioperative mortality rates and obtain a fast and safe management of the aortic rupture. In order to reduce the risks of stent graft infection, we chose a combined technique of stent grafting, esophageal manipulation and mediastinal drainage.

Regarding the therapeutic strategy for closure of the esophageal defect, minimally invasive endoscopic treatment was chosen because the patients were not suitable for surgery.

Because of the minimal size of the esophageal tear and the absence of sepsis, the strategy consisted of placement of a plastic esophageal stent. One week later, after initiation of intravenous antibiotic therapy, the stent allowed drainage of the mediastinal collection that could not be surgically drained.

## 4. Conclusions

Aorto-esophageal fistula is often a terminal event in many patients.

The combined strategy of TEVAR and esophageal endoscopic less invasive treatments was successful in our experience, and it allowed for esophageal healing.

The main limitation of this approach is that it is not as radical as open surgery in reducing infection related to the inability to eliminate the primary infected material.

However, the placement of a plastic esophageal stent allowed the drainage of the mediastinal collection and prevented further contamination of the mediastinum. This, in combination with long-term venous antibiotic therapy, allowed for the complete resolution of the infection and the esophageal healing.

This strategy enabled survival and infection control in patients who, considering initial disease severity and baseline characteristics (such as high average age), were not candidates for surgery.

In the management of aorto-esophageal fistulas, any strategy must be highly individualized, as even the most recent guidelines clearly suggest [5].

A less invasive therapeutic approach represents an alternative in frail patients with high surgical risk, with lower intraoperative and perioperative mortality rates than those associated with thoracotomy.

This combined approach is extremely promising and could represent in the future a more appropriate alternative to the classic open surgical repair.

Certainly, it can only be considered for selected patients, and strict intensive follow-up is required, because infections are often clinically silent.

## Figures and Tables

**Figure 1 jpm-12-01735-f001:**
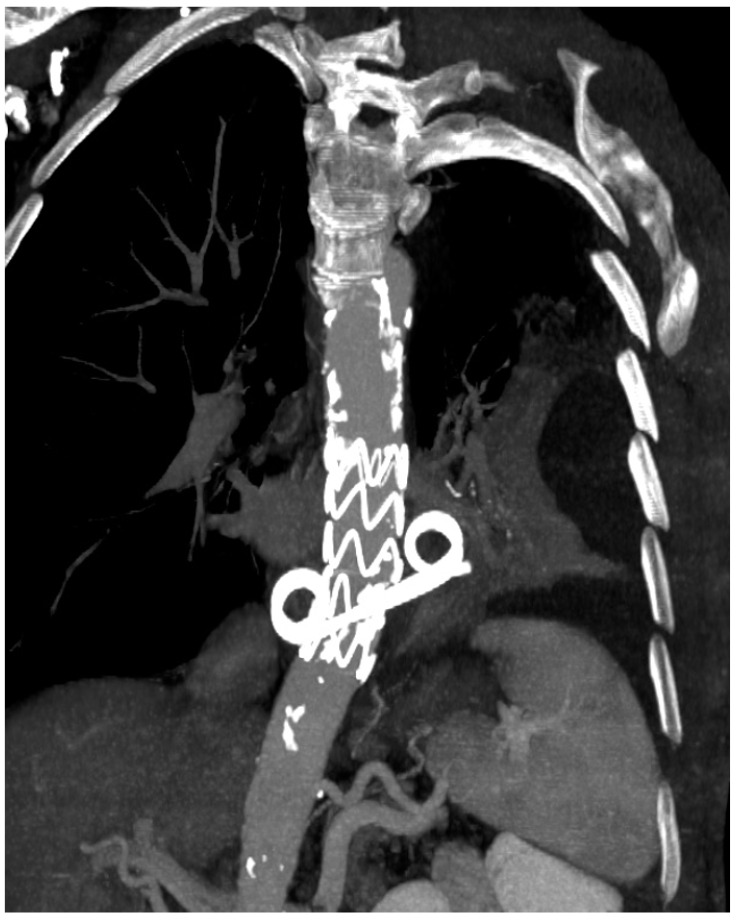
CT showing two pigtail plastic stents in esophagus.

**Figure 2 jpm-12-01735-f002:**
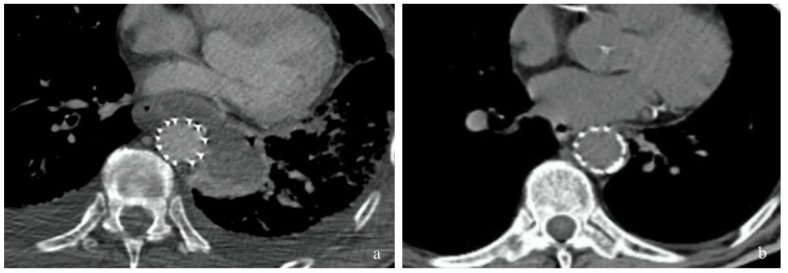
Comparative computed tomography (CT) (**a**) after TEVAR, with evidence of mediastinal collection anterior to the thoracic aorta (**b**) one month after placement of pigtail stent, with evidence of complete reduction of the purulent collection and no signs of mediastinal infection or stent-graft contamination.

**Figure 3 jpm-12-01735-f003:**
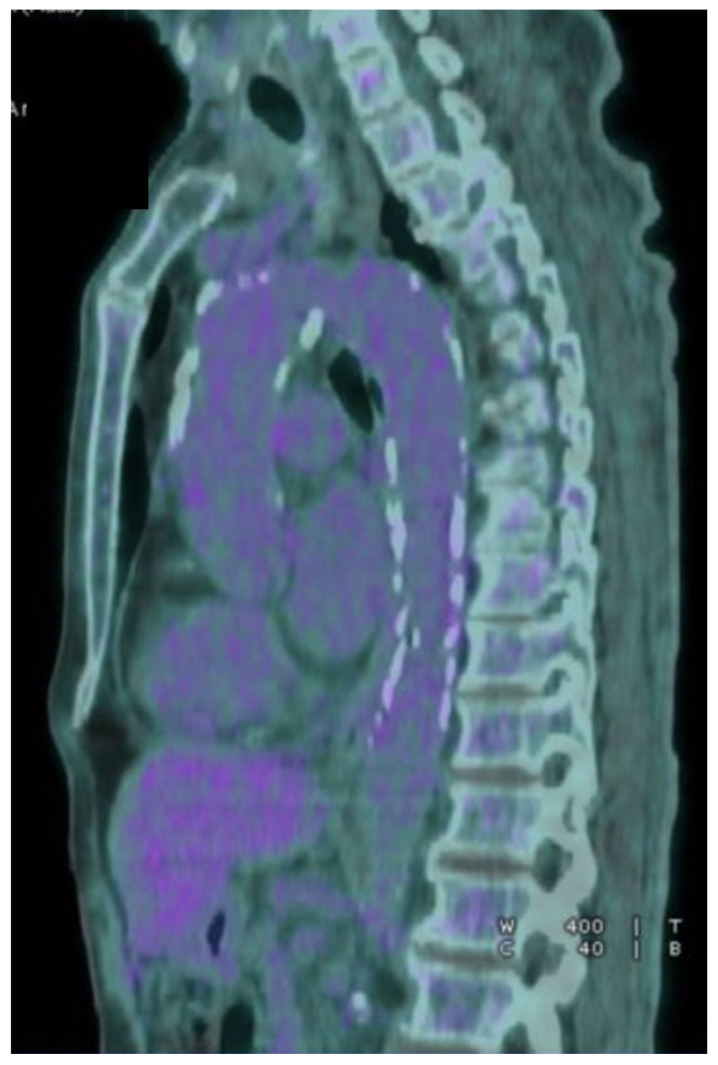
FDG PET-TC upon 6 months of follow-up showed no signs of infections and no leakage.

**Figure 4 jpm-12-01735-f004:**
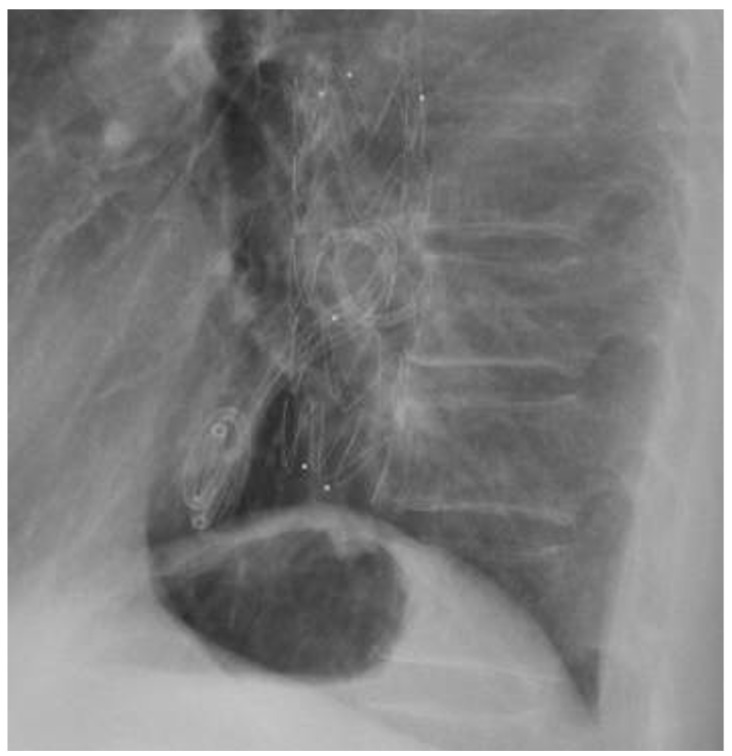
Thoracic X-ray showing two pigtail plastic stents in esophagus.

**Figure 5 jpm-12-01735-f005:**
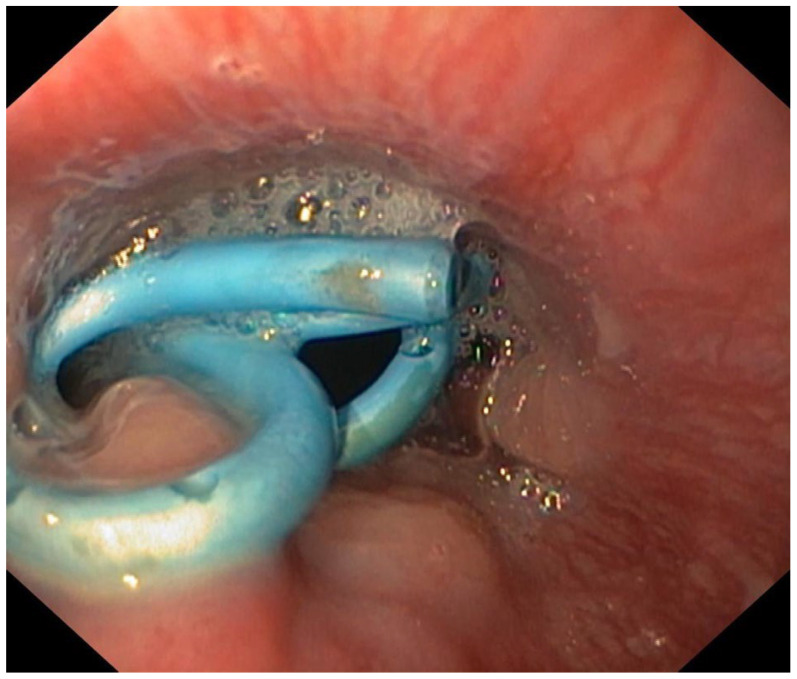
Endoscopy view of pigtail plastic stents.

**Figure 6 jpm-12-01735-f006:**
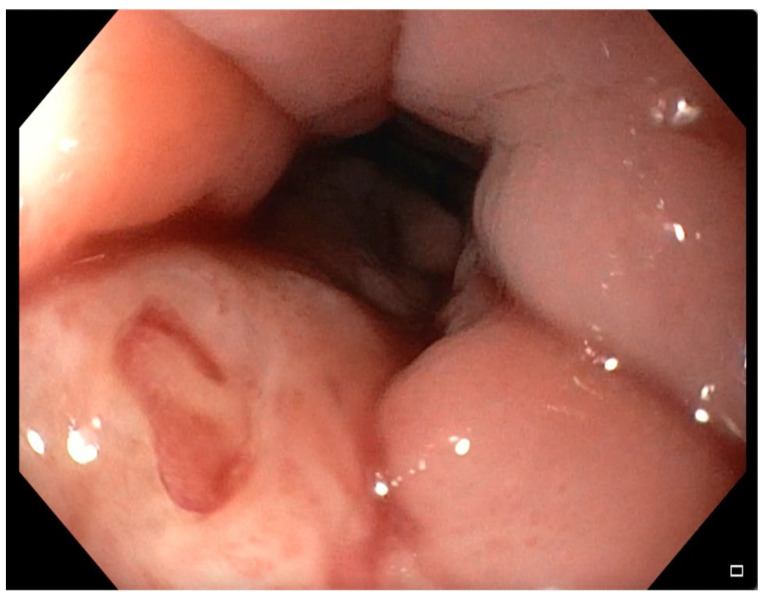
Intraoperative endoscopy evidence of complete healing of the esophageal lesion.

## Data Availability

Not applicable.

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
