# Peer review of "A New Mini-Invasive Approach for a Catastrophic Disease: Staged Endovascular and Endoscopic Treatment of Aorto-Esophageal Fistulas"

_jpm, 2022, doi:10.3390/jpm12101735_

Round 1
Reviewer 1 Report
In the manuscript “Esophageal healing after aorto-esophageal fistula: staged endovascular and endoscopic treatment”. The authors present a case series on the combined strategy of TEVAR and subsequent esophageal endoscopic stenting to recover the AEF.
This is an interesting topic and I appreciate authors’ results on this life-threating problem using a minimal invasive approach (however avoiding major esophageal surgery).
I found this work interesting in its methods and considerations, though I have some comments:
1) The English must be improved. The authors should check the manuscript in order to let it more readable:
Page1, line 40: mellitus diabetes > diabetes mellitus
Page 1, line 42: Laboratory analyses > laboratory test
..
2) Introduction, and particularly Conclusion paragraphs should be improved with more content that point out Authors’ opinion on the argument. What do they think could be the limit of this technique? What is the advantage?
3) I suggest the authors to rephrase the title choosing a stronger one
Reviewer 2 Report
Very interesting case overview. we are looking forward to the long-term results as the majority of patients are no longer alive after 1 year as fas as we know so far. Pls discuss Nabil Chakfé et al, European Society for Vascular Surgery (ESVS) 2020 ClinicalPractice Guidelines on the Management of Vascular Graft and Endograft (Eur J Vasc Endovasc Surg (2020) 59, 339e384) and its algorithm.
I
Round 2
Reviewer 1 Report
I congratulate with authors for the final version of the manuscript.
No further comments.